:::◌: PLOS | ONE

# Conceptualising alcohol consumption in relation to long-term health conditions: Exploring risk in interviewee accounts of drinking and taking medications

**Mary Madden** [1]*, **Stephanie Morris**[1], **Duncan Stewart**[1], **Karl Atkin**[1], **Brendan Gough**[2], **Jim McCambridge**[1]

**1** Department of Health Sciences, Seebohm Rowntree Building, University of York, Heslington, York, United Kingdom, **2** School of Social Sciences, Leeds Beckett University, Leeds, United Kingdom

* mary.madden@york.ac.uk

## Abstract

### Background

Alcohol use is a major contributor to the burden of disease, including long-term non-communicable diseases. Alcohol can also interact with and counter the effects of medications. This study addresses how people with long term conditions, who take multiple medications, experience and understand their alcohol use. The study objective is to explore how people conceptualise the risks posed to their own health from their concurrent alcohol and medicines use.

### Methods and findings

Semi-structured interviews were conducted with a sample of 24 people in the North of England taking medication for long term conditions who drank alcohol twice a week or more often. Transcripts were analysed using a modified framework method with a constructionist thematic analysis. Alcohol was consumed recreationally and to aid with symptoms of sleeplessness, stress and pain. Interviewees were concerned about the felt effects of concurrent alcohol and medicines use and sought ways to minimise the negative effects. Interviewees associated their own drinking with short-term reward, pleasure and relief. Risky drinking was located elsewhere, in the drinking of others. People made experiential, embodied sense of health harms and did not seem aware of, or convinced by, (or in some cases appeared resigned to) future harms to their own health from alcohol use. The study has limitations common to exploratory qualitative studies.

### Conclusions

Health risk communication should be better informed about how people with long-term health conditions perceive health outcomes over time, and how they adopt experience-based safety strategies in contexts in which alcohol consumption is heavily promoted and weakly regulated, whilst medicines adherence is expected. Supporting people to make

**Data Availability Statement:** The study is part of a publicly funded research programme and analyses qualitative interview data for which consent was

obtained to use excerpts in reports and publications but not to make full transcripts available. Interviewees only consented to their data being looked at by members of the research team and, 'data collected during the study may be looked at by responsible individuals from the University of York, the NHS Trust or from regulatory authorities, where it is relevant to my taking part in this research.' The transcripts contain a considerable amount of contextual data, from which it may be possible to identify a participant, but without which the transcripts are not intelligible. Public availability of all relevant data would therefore compromise patient confidentiality and participant privacy. All unrestricted quotations are contained within our manuscript and requests for further access to the remaining minimal data set would need to go through Yorkshire & the Humber - South Yorkshire Research Ethics Committee. As there was no informed consent to make the interview data fully publicly available, requests for data access would need to be considered by the research ethics committee who granted approval for the study. Yorkshire & The Humber - South Yorkshire Research Ethics Committee, Room 001, Jarrow Business Centre, Rolling Mill Road, Jarrow, Tyne & Wear, NE32 3DT REC ref 17/YH/0406; nrescommittee.yorkandhumber-southyorks@nhs.net.

**Funding:** This research was funded by the National Institute for Health Research (NIHR) Programme Grants for Applied Research (RP-PG-0216-20002). The views expressed are those of the authors and not necessarily those of the NIHR or the Department of Health and Social Care. No funding bodies had any role in study design, data collection and analysis, decision to publish, or preparation of the manuscript.

**Competing interests:** The authors have declared that no competing interests exist.

active and informed connections between medicines, alcohol and potential personal health harms requires more than a one-way style of risk communication if it is to be perceived as opening up rather than restricting choice.

## Introduction

Alcohol and pharmaceuticals are health-impacting commodities which are significant in the global economy and incur considerable costs to health services [1, 2]. The use of both is determined by many actors, including national governments, regulatory agencies, industry actors and health systems, as well as health professionals and individuals within their own particular lifeworlds (i.e. the self-evident world of their own lived experience) [3]. Alcohol is a psychoactive drug (ethanol) which presents a major public health challenge [4]. It is currently consumed and understood as a recreational rather than as a pharmaceutical drug, although alcohol has historically been assigned therapeutic value and used as a medicine [5]. Alcohol consumption is deeply culturally ingrained and, in the UK, is commonly associated with enjoyment, relaxation and celebration. Alcohol is also linked to poor health in various and complex ways and presents some risk to all those who use it [6,7].

Although drinking prevalence among adults in England has fallen over the last ten years, those aged 55–64 are most likely to be drinking at hazardous levels [8], meaning with increased risk of chronic physical and mental health morbidities, including gastrointestinal disorders (especially liver and pancreatic disease), cardiovascular diseases, depression, anxiety and cancers or acute intoxication-related injuries or violence [9]. This age group are also more likely than younger age groups to have existing long-term health conditions and take medicines. Many have multimorbidity (several long-term health problems) for which they are taking multiple medicines [10].

Treatment regimes for multiple conditions involve potential for adverse medication interactions and poor adherence [11], and there are growing concerns about potential harms from polypharmacy and the burden for patients and health professionals of managing multiple treatments for long-term conditions [12]. There are also concerns about prescription drug dependence [13]. There is a lack of consensus on what constitutes an alcohol-interactive medicine, but it is known that older adults are particularly vulnerable to adverse effects from concurrent alcohol and medication use [14]. In light of the growing evidence of risk, the UK Chief Medical Officers have set guidelines for low risk drinking at 14 units (where 1 unit is 8 grams of ethanol) of alcohol per week for men and women, spread evenly over three days or more [15]. The aim of the guidelines is to enable people to make informed choices about their alcohol intake.

There is a burgeoning literature on how people living with long-term conditions develop their own expertise in their condition and its management [16]. These include studies of lay epidemiology, i.e. how people individually and collectively assess their risk of illness by linking cases known to them with the circumstances surrounding the illness event [17]. There is some work on the lay epidemiology and pharmacology of medicines [18–20], and Lovatt and colleagues have explored how drinkers interpret alcohol guidelines in the context of their own drinking practices and risk perceptions [21]. Participants in the Lovatt study saw UK drinking guidance as not relevant to their own drinking [21]. This finding was also highlighted by Khadjesari and colleagues' study of primary care patients' views on their own drinking [22]. Authors of these studies identified comparatively little research on understanding patient and

public perspectives on risk communication in comparison to the perspectives of health professionals. A literature on the risks of concurrent medicines and alcohol use among older adults is also developing [14]. This includes a recent qualitative study of mostly recovering dependent drinkers in the North of England, which found complex relationships between alcohol and medicines use which were not discussed with health professionals [23].

This paper contributes to the under-researched area of how people with long term conditions, which involve the use of medicines, experience and understand their use of the psychoactive drug, alcohol (ethanol). It reports on findings from a qualitative study which formed part of the intervention development phase of a five year research programme to develop a basis for discussion of alcohol within community pharmacy- based medication reviews. An earlier report found that inquiries about drinking from health professionals were usually experienced as referring to problems in the form of dependence on alcohol rather than about the possible consequences for long-term health [24]. This paper explores in more depth how people conceptualised alcohol in relation to their own health and medicines use. Findings are discussed in the context of the recent public health call to more clearly recognise alcohol as a drug [4], and the conclusions drawn in a review of qualitative studies of alcohol consumption among older adults [25].

## Method

Interviewees were recruited to a study focusing on medication use, drinking and the appropriateness of discussing alcohol consumption with community pharmacists. The study received NHS Health Research Authority research ethics approval (Yorkshire & the Humber—South Yorkshire Research Ethics Committee REC reference 17/YH/0406). Two researchers, a sociologist (MM) and an anthropologist (SM), conducted semi-structured interviews with participants in their own homes using a topic guide, which comprised open-ended questions. Semi-structured interviews were chosen because they allowed for flexibility in data collection and produced rich narratives about perceptions and experiences which permitted analysis of how participants made sense of the topic under investigation.

The interview schedule and recruitment materials were both developed with the input of lay advisors as part of the research programme commitment to co-production and patient and public involvement (reported elsewhere). This was also consistent with the funder expectations. Recruitment was pragmatic, aiming for a diverse sample of people potentially eligible for medication reviews who drank alcohol twice a week or more often within the limitations of the field work time frame (March–April 2018). Eligible groups for medication review at this time included those: taking high risk medicines, including non-steroidal anti-inflammatory drugs (NSAIDs), diuretics, antiplatelets and anticoagulants; recently discharged from hospital where changes had been made to medicines; with respiratory disease; at risk of or diagnosed with cardiovascular disease; newly prescribed medication for type 2 diabetes, asthma and chronic obstructive pulmonary disease (COPD), hypertension or antiplatelet/anticoagulant therapy [26, 27].

Recruitment was conducted in seven community pharmacies which offered medicines reviews on behalf of the NHS in three locations in the North of England, in neighbourhoods with different levels of relative deprivation as measured in the English Index of Multiple Deprivation (see Table 1) [28]. Following the categorization of pharmacy ownership used by Bush et al. [29], six of the seven pharmacies were large chains and one was a small chain.

Researchers (SM and MM) encouraged referrals through pharmacy staff and attended pharmacies at different times of day to recruit directly. The number of people approached directly was not systematically recorded. Posters and flyers about the study were displayed, placed on pharmacy counters and distributed to people waiting for their prescriptions. These

**Table 1. Types of pharmacies where participants were recruited.**

|  | Type | Location | IMD Decile (1 = most deprived, 10 = least deprived) |
|---|---|---|---|
| Pharmacy 1 | Large Chain | Small City | 7 |
| Pharmacy 2 | Large Chain | Large town | 1 |
| Pharmacy 3 | Small Chain | Village | 10 |
| Pharmacy 4 | Large Chain | Town | 2 |
| Pharmacy 5 | Large Chain | Town | 4 |
| Pharmacy 6 | Large Chain | Small City | 3 |
| Pharmacy 7 | Large Chain | Small City | 3 |

asked: "Do you take medications long term? And do you drink alcohol twice a week or more?" If people answered yes to the flyer questions and were interested, pharmacy staff or a researcher provided brief information about what was involved and asked for a telephone number. A researcher then called potential participants to confirm their interest in the study and provide further information.

People willing to be interviewed were asked to confirm drinking status using the shortened three item form of the Alcohol Use Disorders Identification Test (AUDIT-C) [30], and medication adherence using the five item Medication Adherence Report Scale (MARS) [31]. The aim was to recruit a range of participants with low and high levels of medication adherence and a 'positive' AUDIT-C score (here using 4/3 thresholds for men and women respectively). Generally, the higher the AUDIT-C score (0–12), the more likely it is that drinking is affecting the person's health and safety [30]. A high MARS score (on a scale of 5–25) indicates a high level of self-reported medicines adherence [31]. These measures were used to assess eligibility and to give context to interviewee responses. Eligible patients were sent a recruitment pack which included an invitation letter, a participant information sheet, and a study consent form. Contact details for the research team were also provided to enable participants to ask any further questions in advance of the interview. Written consent was obtained at interview and participants were given assurances about their rights, confidentiality and anonymity of their responses. Participants received a £10 shopping voucher to thank them for their time.

Interviews were digitally recorded and transcribed. A modified framework method was used to organise and present data from transcripts [32]. This supported a constructionist thematic analysis which recognised the constitutive nature of language without focusing on micro level language use [33]. NVivo 11 software was used for data management and coding. Some a-priori codes were established from the interview schedule, but detailed coding was developed in analysis rather than predetermined. The stability and reproducibility of the coding was checked within the broader co-investigator team by comparing co-investigators' preliminary analysis of sample scripts with those of the main field researchers. The analysis presented below focuses on concurrent medicine and alcohol use. The first author led on this analysis and it was developed iteratively through presenting drafts to the full author team for feedback. Participants have been anonymised and given identifier codes which convey some simple description. For example, INV-015-M-24-05 means that interviewee number 15 is male with a MARS score of 24 and an AUDIT-C score of 5.

## Results

### Demographic data

A total of 43 people expressed an interest in the study during the recruitment period; 39 provided telephone numbers to researchers in pharmacies, two were referred by pharmacists and

two responded independently to posters and leaflets. Interviews were conducted with a sample of 25 people selected based on eligibility and availability within the timeframe. One of the interviewees was drinking at harmful levels (AUDIT = 12) and taking just one prescribed medication (methadone) but had no long-term condition other than substance dependence, so has been removed from this particular analysis. The resulting sample of 24 contained fewer women than men and had little diversity in terms of ethnicity or sexual orientation (see Table 2).

## Alcohol in the mix with medications

Interviewees had positive AUDIT-C scores ranging from 3–12 (mean = 7.88) and mostly described themselves as "casual", "social" or "moderate" drinkers. 12/24 (50%) had AUDIT-C scores ranging between 8–11, and one man had a score of 12. He and two other interviewees (12.5%) had received treatment for drug and/or alcohol dependence. 17/24 interviewees (71%) said they drank more heavily in the past than they do now. Four (16%) interviewees said their drinking was at a similar level over time, and three (12.5%) said they drank more now than they did in the past. Reasons given for this were having more money, children having grown up and no longer riding a motorbike.

Interviewees had high MARS scores, which meant that they self-identified as medicines adherent (range: 15–25; mean = 23.13). The person with the lowest MARS score (15) had an AUDIT-C score of 11 and in the past had been in treatment for alcohol dependence. Nine other interviewees (37%) mentioned consultations of varying lengths with a health professional where they had been advised they should reduce drinking. Interviewee views on discussing alcohol with health professionals have been reported elsewhere [24]. Some interviewees had a lifetime of managing different health problems, others were feeling well until relatively recently, while others were still feeling well and using preventative medication. Most interviewees reported taking multiple prescribed medications (range 1–16; mean 6.8). The person

**Table 2. Characteristics of the interview sample (n = 24).**

|  | N (%) | Mean (SD) |
|---|---|---|
| Male | 15 (63) |  |
| Female | 9 (38) |  |
| Age |  | 64.13 (12.66) |
| Higher Education | 7 (29) |  |
| Employment |  |  |
| Retired | 13 (54) |  |
| Semi-retired | 1 (4) |  |
| Sickness, disability or unemployment benefits | 6 (25) |  |
| Employed | 2 (8) |  |
| Unknown | 2 (8) |  |
| White British or Irish | 24 (100) |  |
| Heterosexual | 24 (100) |  |
| Partner status |  |  |
| Married or partner | 15 (63) |  |
| Divorced | 3 (13) |  |
| Widowed | 3 (13) |  |
| Single | 3 (13) |  |
| AUDIT-C score |  | 7.88 (2.72) |
| MARS score |  | 23.13 (2.15) |

taking one medication was taking this for epilepsy. 14 (58%) were also taking more than one supplementary online or over-the-counter substances (range 2–6; mean 2). Three of the sample of 24 (12%) were also using illicit drugs, including one who was prescribed opioid substitution therapy and was taking multiple medications for other long-term conditions.

Some of the interviewees were familiar with the names of the medicines they were taking, others used more generalised terms to describe 'tablets' they took or 'inhalers' they used. Some people talked about medicines mostly in terms of the conditions for which they were being taken. Some directly showed the interviewer their prescription or the medicines themselves. The list of prescribed medicines derived from interviews (S1 Appendix) was therefore not exhaustive:

> If you said, what's that yellow one, I don't know. . .

> (INV-004-M-25-07).

The majority (16/24;67%) were taking medicines to counter side effects of others including: proton pump inhibitors and H2 (histamine-2) blockers to reduce stomach acid (9/24, 37%); laxatives (5/24, 21%); diuretics (2/24, 8%); medication for leg cramps linked to statins (1/24, 4%) and anti-emetics (2/24, 8%). One man taking medication to relieve swollen ankles and a cough from gastric irritation caused by other medications said:

> There are side effects . . . with all medicines it's swings and roundabouts. . . A free problem with everything

> (INV-015-M-24-05).

Interviewees were taking medicines involving most of the 38 potentially serious alcohol-medication interactions in older adults according to the POSAMINO criteria recently identified by Holton et al [34]. These included 13/24 (54%) taking medicines for cardiovascular disease; 6/24 (25%) taking non-steroidal anti-inflammatory drugs for musculoskeletal and joint diseases and 2/24 (8%) taking medicines for diabetes. 18/24 (75%) were taking medicines which impact on the central nervous system including antidepressants (6/24, 25%), opioids (5/24, 21%) and benzodiazepine-related medications (3/24, 12.5%), the widespread use of which has long been the subject of public health concern in the UK because of the potential for dependence [13]. Medicines taken by people in this sample had the potential to interact with alcohol in ways which can result in adverse and non-trivial health effects for the drinker, including sedation, hypotension, gastrointestinal bleeds, hypoglycaemia and liver damage.

Interviewees described regular routines and ways of storing medicines that helped them to remember to take their medicines. They also described regular drinking routines. Except for two of the heaviest drinkers, most sought to separate their medicines consumption from their alcohol consumption, even if the interval was brief:

> I have like a Tupperware tub with the stuff in and I usually leave that out on the back of the worktop, sort of to remind me, say, if I go back in the kitchen after I've had my breakfast, it's there to remind me to have it. But, you know, occasionally I have forgot . . .. if we were out and we're staying out, you know, having a drink afterwards, I would take my tablets sort of just as I'd finished eating or as I was eating, and then maybe wait another half hour or so before I had a drink. I don't tend to drink while taking the tablets

> (INV-019-F-21-9).

**Alcohol-related adherence breaches.** Alcohol consumption has been associated with poorer medication adherence [35, 36], but most people in the sample, including heavier drinkers, had high MARS (adherence) scores. During the course of their interviews five people recalled other times when they had forgotten or skipped their medication which had not come to mind when answering the MARS screening questions. They reported these as exceptions to their usual adherence. Similarly, as people talked in their interviews and began to elaborate on their patterns of alcohol consumption, they described nightcaps, top ups and special offers which meant they drank more alcohol than they had initially said. They also recalled disruptions to their routine, like going on holiday, a special occasion or when friends were visiting, which meant they drank more, or their medicines routine was disrupted. For example, a woman aged 48 with the maximum MARS adherence score of 25 and a high AUDIT-C score of 10 recalled forgetting to take her medications after drinking too much:

> I've probably forgotten to take them twice, maybe . . . I was drunk . . . I've just fallen into bed and forgotten all about it . . . I think it's only happened twice, as far as I can remember, and I just skipped it. I just took them the next day . . . I just thought, that's really stupid, and you did drink too much and that's what's happened . . . I kind of thought, it's just 24 hours. It will be okay

> (INV-006-F-25-10).

A man aged 57 with and AUDIT-C score of 10, who was taking multiple medications including an opioid for severe arthritis, thyroxine and medications for gout and anxiety, had difficulty recalling whether he had missed taking his medications and attributed any forgetfulness to worries rather than drinking:

> I might have done . . . sometimes, not the drinking but other things in my head, if I'm worried about my daughter or something, have I had my medication again, and I look at the pack. I know I don't take [it] every day

> (INV-012-M-23-10).

Similarly, another man with a high AUDIT-C score (11) aged 69 who had received treatment for his alcohol use in the past did not attribute his forgetfulness to drinking, although he was drinking throughout the day and had trouble managing multiple medications, "it's not 'cause of the alcohol. . .I'm very forgetful" (INV -018-M-23-11).

A woman aged 75 with a relatively low AUDIT-C score (3) and high medication adherence score (MARS = 22) reported deliberate non-adherence to her statin medication. She explained she was concerned that her dose was too high compared to other people she knew. She had not discussed her concerns with her doctor or pharmacist but had made her own decision to take only half of the dose she was prescribed by taking her statin every other night. In addition, she deliberately skipped her statin altogether when she had been out drinking:

> . . . if I have been out and had a few drinks I'm very reluctant to come home and have . . . I think, oh, no, I'll not take my statins, I've had a drink

> (INV-013-F-22-03).

Although she did not want to mix the statin with alcohol on these drinking occasions, she regularly had a whisky at night to help her sleep.

**Contrasting concerns about side effects and taking drugs over time.** Drinking alcohol and taking medication were health-impacting drug consumption practices that were viewed differently in respect of self-image, risk and autonomy. Alongside perceived benefits, there was some reluctance to take medication to improve health because of potential hazards, with this reluctance regarded as the exercise of personal control. This contrasted sharply with perceptions of alcohol use, where the hazards were minimised, or the use of this drug was defended as an exercise in personal freedom.

Many people said they did not like taking long-term medication, seeing it as a sign of ageing:

I suppose my only worry is that sort of old age one of I don't like taking tablets . . . my mum used to have loads of tablets every day . . . in like little containers, and I just thought, oh God, I'm going to be like that; so just that, the sort of stigma attached to it I suppose . . . It's kind of a, I wish I didn't have to but I know I do, kind of I'm resigned to it now. . . I've got to take them

(INV-006-F-25-10).

Others, including some of the heaviest drinkers, were concerned about possible unforeseen consequences:

I'd rather not be taking long-term medication because there's always possibilities of side-effects of taking it long-term that hasn't come to light or that might affect me. So there's always a concern about that but there's nothing I can do about it . . .

(INV-005-M-25-11).

Taking medicines over time was generally viewed as potentially risky:

They [my children] say, you're taking them too long . . . One of them [an antiarrhythmic] is giving me something with my liver now but the doctor said . . . it's there for another reason and that's a minor thing . . . they're all bad for you. When you look at the television or . . . they all have side effects, and you get a paper this big telling you, . . . Well you wouldn't ever take one if you read what it was supposed to do. So, you just take their word for it

(INV-004-M-25-07).

. . . obviously, I don't know what . . . it's doing to my insides, but then without them, I probably wouldn't be here. So it's sort of a no-win situation. If I take them, I'm probably doing damage to my liver and my kidney. But on the other hand, without them I could be dead

(INV-016-F-23-04).

Some had refused medicines or expressed reluctance to take them because of potential side effects and were not sure if their medications were working. This man had refused statins and talked about stopping his blood pressure medicines as the only means of determining their effectiveness. Taking medicines conflicted with his lifelong view of himself as healthy:

Statins. I said, no, I've heard too much about people who have had adverse effects . . . people say, are they doing you any good, and I always say, well, I don't know, they might be and the only way I find out if they weren't doing me any good was stop taking them. And if I

have any effects then, adverse effects, I'd go back on the tablets . . . I've always thought of myself of being fit and I didn't need all this stuff. But now, I mean, it's [over the counter medicines to reduce stomach acid] and your tablets in the morning

(INV-002-M-25-06).

Many of the interviewees had experienced unpleasant side effects from medication and some found it difficult to distinguish the feeling of a side effect, adverse event or symptoms of withdrawal from a symptom of their conditions, or an effect or after-effect of drinking alcohol:

I just don't notice them [side effects], I think. . ..I've got that many things wrong with me, I just don't notice them

(INV-007-M-15-11).

**Drinking despite some knowledge of personal health risks.**   During the course of the interviews 13/24 people (54%) acknowledged alcohol as posing some potential threat to their own health, or that drinking should be avoided with their medication. For most of these people, this was not enough to alter their consumption. A woman with a high AUDIT-C score of 11, aged 41, taking multiple medications including antidepressants, said she felt guilty after drinking because she knew it could be compromising her treatment, but at the time it was worth it for the benefits it bestowed:

I'm worried about conceiving but I'm also worried that it might not be helping my depression really to be drinking . . . Because how is the medicine going to help me if I'm depressing myself with alcohol? . . . you listen but whether you follow the advice is another thing. And I don't see much harm . . . three glasses is not going to kill me . . . If I was drinking more I think I'd be a bit worried . . . it seems okay to me, so . . . and I don't get pickled . . . I know it's not advised. . . but sometimes you just . . . want a drink so . . . they said that if I must drink . . . keep it to a minimum and try not to mix as well. And I said, well I must occasionally drink, so yes, I will do as I'm told. I will take it steady. So I never get drunk, drunk. I'll just have a few glasses, get a little wine glow and go to bed. . . I feel guilty as hell after I've had a drink, yeah. Yeah, I feel bad. I think, oh you're not helping yourself. But at the time I'm having the drink I'm thinking, I'm going to sleep well tonight . . .

(INV-017-F-24-11).

Here she spoke about feelings of guilt from "not helping" herself by drinking but also the pleasures of, "a little wine glow". She was drinking regularly at levels higher than recommended in the UK guidelines and had been advised by health professionals to, "keep it to a minimum". She wanted to 'do as she was told' and had interpreted "minimum" as taking it "steady". She said she did not see much harm in drinking three glasses of wine regularly at night. She said she had identified her limit from the experience of feeling bad when she had exceeded it: "If I drink more than that I get pickled and then I'm not fine . . ." (INV-017-F-24-11).

This woman was one of a number of people in the sample who said they used alcohol to aid sleep. Many interviewees complained about difficulties sleeping, including disruption because of night-time urination caused by conditions or medication. Alcohol is a diuretic which can speed up urine production. It also disrupts sleep in other ways [37], however only one person said they should probably reduce their alcohol intake because of sleep disruption. Perceptions of the effectiveness of drinking to help sleep were nevertheless mixed. Some used alcohol to

help them to go to sleep instead or on top of medication, but also said they woke up a few hours later. For example, one woman said:

> I only have it [whisky] to make me sleep . . . it makes me sleep . . . I go straight to sleep but it's only for a few . . . as my friends said, drink, you might stay asleep for a few hours but . . . won't . . . sleep for eight hours or whatever
>
> (INV-013-F-22-03).

The woman mentioned previously with the high AUDIT-C score of 11 said she was worried about alcohol compromising her health, but she was more worried about using sleeping tablets than alcohol because they might be addictive. She described alcohol as effective and less of a risk, especially as she did not experience any negative after-effects:

> . . . the drinking knocks me out . . . I really do sleep . . . I'm desperate for a method that will make me sleep without any alcohol needed or anything else . . . I just worry about how addictive they [sleeping pills] might be because I know that prescription ones a lot of people get addicted to . . . I don't want to risk it . . . Most people should suffer after three glasses of wine the next day but I don't
>
> (INV-017-F-24-11).

Another woman with a high AUDIT-C score (9), aged 47 and taking medication for anxiety, depression, mitochondrial disease and acid reflux, also said she knew alcohol was not good for her, but she enjoyed the "relief" it brought:

> . . . they do say alcohol probably won't help my condition, although, yeah, it feels like it helps now and again, it's just that relief
>
> (INV-019-F-21-9).

Some interviewees said that although they were aware of the risks of drinking alcohol in relation to their medicines, they ignored them, including this woman with a high AUDIT-C score of 10:

> It's on all of them . . . I think especially the Sertraline it says avoid too much alcohol, or words to that effect, and I think the Atorvastatin as well . . . I presume it's because the effects of the alcohol will be multiplied or heightened in some way . . . I've never thought about that. I just know that I shouldn't, but I do. . . I suppose I am aware that my hangovers are worse since starting on some meds, so I do tend to cut off more in the week ones [drinking sessions], in the work next day ones. But no, I just ignore it mostly
>
> (INV-006-F-25-10).

She described an interesting mixture of risk awareness combined with a disregard of that risk in her responses. There was less ambivalence in this man's pithy rejection of risk communication:

> . . . I've been told it [alcohol] makes you more fatty on your liver and all that, but . . . I don't want to live forever anyway, so bollocks . . . No one's ever talked to me about [mixing

alcohol and medications] . . . I've told the doctor I drink. They say the same as me, everyone's got their own vice . . . I'm happy with the risk, I'm happy as Larry

(INV-012-M-23-10).

This man aged 57 with a high AUDIT-C score of 10, acknowledged risk in terms of the potential of alcohol to shorten life but not the possibility of risk related to medicines, because in his experience medicines did not "affect the alcohol". His medication included an opioid which carries high risk of interaction, as well as medications for cardiovascular disease, gout, anxiety and post thyroidectomy. His risk focus was on alcohol accelerating mortality rather than the potential of alcohol to diminish his health-related quality of life. This included pain from gout due to the build-up of uric acid exacerbated by alcohol of which he seemed unaware [38].

Another man aged 66, with an AUDIT-C score of7and taking medicines for prevention of cardiovascular disease, said that he sometimes considered the nature of the risk and had 'told himself' it was small:

I wouldn't say I've never thought about it [impact on blood pressure and cholesterol], of course I have, you know, it does cross mind, it does cross mind. But I suppose that I've told myself that the amounts that I have and the conditions that I have, it won't make an awful lot of difference

(INV-022-M-24-7).

Similarly, other interviewees made judgements about the extent of the risk involved, sometimes with the aid of health professionals. A man aged 67 taking similar medication and drinking at high, levels (AUDIT-C = 11) had sought reassurance from talking to his doctor and from his blood tests that his drinking was okay:

I know that a lot of the medication says, don't take this if you drink heavily or. . . [it] doesn't really define that, which is why I chatted to the doctor about it saying, is it a problem me taking this amount, and he says, no. So no, there's nothing that sort of gets in the way of each other. . . I also like to have a blood test fairly often to make sure that everything is still okay given that I take these drugs and given that I drink as well which I know is not a good combination necessarily for these

(INV-005-M-25-11).

The doctor is not reported as mentioning the potential impact of alcohol on the effectiveness of his medicines for high blood pressure and high cholesterol or as a gastric irritant in combination with aspirin. In terms of clarifying the meaning of "drink heavily", he says the doctor said:

. . . it's all around food and that's better than going down the pub and downing half a dozen pints of beer. He said he can't see too much wrong with it even though it's above recommended

(INV-005-M-25-11).

Health professionals were represented in some interviews as inclined to tell people not to drink or to cut down, or as in this interview, not wanting to exclude people from socially

acceptable patterns of drinking. A woman aged 41 with trigeminal neuralgia explained how she had started to drink again, while being mindful of risk, after talking to her consultant on New Year's Eve:

> . . . quite a lot of the pain meds, especially the controlled drugs, it says do not drink. So I didn't drink at all. And then I went to see my pain specialist at the hospital and it was New Year's Eve, and he said, are you going out tonight? And I said, well, no, there's no point, I can't drink, you know, and I'm tired. And he said, what do you mean, you can't drink? I said, well, on the meds it says don't drink. And he said, well, you can have a drink, it's just that the effects of the alcohol will be far greater than if you weren't taking them . . . so you've just got to adapt your body and some people are more affected, and some people are less affected. So I'm definitely more . . . if I have a drink, I'm more mindful of, one, what I'm drinking, and, two, how much I do drink. . .

> (INV-016-F-23-04).

Some, including this man aged 79 taking preventative cardiac medications, talked about the difficulty in making sense of the information to gauge the personal risks from alcohol and "make a reasoned decision":

> I think I'm aware of damage that it can do if you drink regularly and heavily . . . it appears sometimes that some of the information is bordering on the scare-mongering . . . where they talk about units and . . . you go along for a while with the accepted limit of one glass with a meal, fine. And then somebody else comes along and says, oh no, no, no, only half a glass and . . . that small amount, that difference, yes, it can be significant, I accept that . . . But . . . and then someone else will come along a week later and say, it can be beneficial to have a glass of red wine with a meal. So you are weighing up these different levels of information and you are saying to yourself, right, I need to have an intelligent norm somewhere in between all that and you sort out the information like that . . . I feel that I'm able to make a reasoned decision

> (INV-015-M-24-05).

Two people aged 48 and 54, with high AUDIT-C scores (11 and 12) were long-term users of illicit drugs and medication, including opioids, hypnotics and antidepressants and were at very high risk of potentially serious alcohol-medication interactions. They were the people in the sample who were most aware of the potential risks to their own health:

> I shouldn't be drinking on them, purely and simply. The doctor would go mad. . . being controlled drugs . . . one of my coping mechanisms I guess, just to block everything out

> (INV-007-M-15-11).

> If you go onto a different medication, they'll instantly say . . . All of them. . .the diazepam, I'm definitely not [supposed to drink], but that's the buzz that I want. . .I'm sort of aware, I know how far I can go with it. I am very cautious, basically. Because I've lost a lot of friends . . . [who] died because of drinking and tablets

> (INV-010-F-22-12).

They said they used drugs and alcohol to socialise and to block out their concerns, or sought intoxication effects, in some cases having regard to risk.

**Locating risky drinking in the drinking of others.** A sense of self-control in relation to drinking, limited the sense in which alcohol was regarded as posing health risks for some interviewees. Most of the 11/24 interviewees (46%) who said that alcohol was not a risk to their own health also said that would change if their consumption increased. A man aged 61 taking medicines for epilepsy with an AUDIT-C score of 8 said:

> Never had any side effects or anything [from drinking], if it got out of control maybe . . . I know for a fact I'm nowhere near that stage
>
> (INV-003-M-24-08).

There was a sense of exceptionalism in some accounts, which attributed susceptibility to harm from the effects of alcohol to those drinkers who drank more than the interviewee themselves or to those who were pre-disposed to problems with alcohol. The reported excesses and susceptibility of others provided a basis for their own restriction logics:

> . . . it's never been what I call a health problem, like my brother . . . in a way it killed my brother, he was alcoholic . . . he died of a heart attack but it was the alcohol . . .
>
> (INV-002-M-25-06).

> I think if I excessed over the eight [cans], whoa, slow down there. I know that myself . . . It's like he [neighbour upstairs] can get up in the morning and open a can. I could never do that. I would balk at the idea . . . Physically it would make me balk
>
> (INV-012-M-23-10).

Regulating drinking by consuming a consistent amount regularly was represented in a number of interviews as safe, as long as this was not exceeded. At the same time, people said that those with problems don't always know they have a problem:

> I've got this awful assumption that I think I'm fine, but isn't that one of the problems with people who've got problems, that they often don't know they've got the problem?
>
> (INV-020-F-23-9).

> If I thought I had a problem . . . people with problem drinking tend not to realise it, so they never ask for advice . . . I'd like to think I'd have the common sense to cut back if I thought I was having a problem
>
> (INV-021-M-24-7).

Again, there was some personal exceptionalism and 'othering' in this expressed hope that "common sense" would pertain in a circumstance where it was thought to be absent in others.

**Experience-based safety strategies.** Personal risks from combining alcohol and medications were largely judged in relation to short-term bodily experience rather than longer term or systemic effects. A woman taking multiple medications, including an antidepressant, a hypnotic and medication for severe pain, high blood pressure and acid reflux, said that she delayed taking her antidepressant when she has a drink because she knew from past experience that she would fall asleep at the table, "which is not a good look" (INV-016-F-23-04). Other people

described similar learning-by-experience about the effects of their medication in relation to alcohol. For example, this woman with a high AUDIT-C score of 11, spoke about past experiences of drinking on antidepressants:

> I know I couldn't drink when I was on the Seroxat, the Paroxetine, because one drink and I'd be, whoa, my head would go, you know, and, no, it weren't nice to get drunk on one drink and stuff . . . The side effects weren't very nice and I would get raging trots the next day after a drink if I had a drink on that, so I didn't drink on the Paroxetine
>
> (INV-017-F-24-11).

Unless they were too severe or got in the way of their lives, people spoke about putting up with the side effects of drinking with medications because of the potential benefits of drinking. The side effects from alcohol were characterised as a short-term price paid for enjoyment. Most people said they used alcohol to relax, to help them to sleep, or as another painkiller for emotional and physical pain. This included, ironically, using alcohol to treat the pain of the hangover caused by overconsumption of alcohol:

> It's just there, I just do it, whether or not I enjoy it . . . It's just there . . . it usually takes the pain away
>
> (INV-018-M-23-11).

Some people noted that their painkillers minimised their hangovers. Some saw this as a bonus but a woman with a high AUDIT-C score (9), aged 62, wondered whether this might indicate that she was taking too much medication:

> I don't suffer the next day like others might, which makes me think, is there a bit of residue of painkiller that works on the hangover as well as everything else, I don't know . . . I don't suffer hangovers, but I'm assuming it's part of . . . do I take too many painkillers? If you're already on a lot of that, there must be a level where . . .
>
> (INV-020-F-23-9).

A man aged 70 with a high AUDIT-C score of 11, said he sometimes felt bad when he took his codeine and had been advised not to drink on this:

> . . . you're advised not to drink with them. But it's not a strong beer, it's mild. It's quite weak. . . . the codeine that's the effect, not the alcohol
>
> (INV-014-M-22-11).

This sense that choosing lower alcohol beer would aid safety was especially common among heavier drinking men in the sample who, after describing their limits, went on to talk about "tots of whisky" or other top-ups:

> They're not strong. It's only four per cent . . . I mean, most lager is 5.2 or something. That does me. That's why I have a bottle of cider. I top it up
>
> (INV-012-M-23-10).

. . .I like to have four pints every night of mild or mix bitter. And it makes me sleep a bit better. . . I'm just happy with that. I could drink more but I wouldn't be happy. . . So, no, I feel great. . . Five pints, top. But I know when I've had enough . . . I have a tot of whisky, not every night . . . And that's with hot water normally

(INV-014-M-22-11).

There were a range of strategies to minimise any damage from alcohol by diluting it or flushing it out of the system:

I try and flush myself out with water the next day. I drink more water and like clear my system . . . I know that it helps flush out the alcohol from your body and I don't want to be just pickling my liver

(INV-017-F-24-11).

. . . I always have a couple of pints of water a night, over the night. And I drink water all day long

(INV-014-M-22-11).

Many of the interviewee shared a sense that safety came by setting drinking limits which, though exceeding officially recommended low risk limits, were lower than the excesses of the past or the excesses of others:

Four pints, five pints doesn't affect me. After what I drank over the years . . . I'm happy because I don't feel bad . . . I know when I don't feel right . . .

(INV-014-M-22-11).

Not being able to face a drink after drinking too much was taken by some of the heavier drinking men to indicate there was not a problem with alcohol dependence. Not drinking enough to make yourself feel ill was another indicator of safety for many interviewees:

I don't drink that much to feel ill . . . sometimes I think perhaps I shouldn't have anything to drink because it affects your tablets, doesn't it? . . . I think, oh, you're not supposed . . . you know, and then my friend will say, oh, Peter . . . well, he's died now . . . Peter has a really bad heart but he'd have a drink and all that, you know, but I don't take notice of . . . I try not to take notice of people. I do what I feel's right

(INV-013-F-22-03).

In addition to resisting peer pressure and paying attention to the number or strength of drinks, people had various rationales for choosing an alcoholic drink which would minimise any potential bodily discomforts or negative impacts on sleep:

. . . [I] just experimented and tried it really. I always used to drink really bottles of lager . . .but . . . I get really, really bloated and quite gassy with that now . . . the discomfort far outweighed the enjoyment, and so . . . I sort of like crossed off drinking lagers and ciders . . . and I went onto . . . I tried spirits. And somebody had suggested, well, just stick with a

clear. And I did that and it doesn't seem as . . . the reactions don't seem as bad, so I tend to just stick with that. And champagne

(INV-016-F-23-04).

## Discussion

By exploring lay perspectives on using alcohol and medicines together, this study provides new insights into the way people taking medicines for long-term conditions experience and understand their own drinking. Alcohol's association with short-term pleasure and a sense of "relief" meant there was minimal attention paid by people to the less immediately tangible risks alcohol poses to their own long-term health and mortality.

The World Health Organisation has described the overuse, underuse or misuse of medicines (by health care providers and individuals) as a major global problem in terms of resource waste and health hazards [11]. The more drugs prescribed, the more difficult they become to manage, the higher the potential risks of a drug-drug interaction, an adverse drug event, or non-adherence. Most interviewees had multimorbidity and worked to fit the associated use of multiple medicines into their daily lives, and this included the regular consumption of alcohol. Many were taking medicines for conditions exacerbated by alcohol use and with risk of interaction. However, the domestic and social lifeworld of alcohol consumption was largely separated from the lifeworld of medicines, health and illness.

Alcohol was generally distanced as a threat to interviewees' own health and seen as such only if it was "a problem" manifested in drinking larger quantities than currently consumed and/or in feeling the compulsion to drink [39]. The interviewees appeared to largely accept their current situations, including risks that were present from alcohol and medications, and to have in place strategies for actively managing these. Ideas about acceptable levels of alcohol consumption and 'embodied' forms of risk assessment were determined not by low risk drinking guidelines but through experience of how drinking made them feel [40] [21], including, for some, testing of warnings not to mix alcohol with medicines. People set their own personal limits calibrated by how drinking made them feel. Recall of actual amounts/doses consumed was not always clear or consistent. Personal health risks or harms from drinking were mostly described in terms of short-term, localised, effects such as headaches and stomach aches, rather than any cumulative or long-term, systemic effects of taking a potent psychoactive neurotoxin. Interviewees in the study were 'delay discounting' in the sense that they were more invested in positive rewards from drinking alcohol in the present, e.g. getting them to sleep, and in managing the short-term risks, than any positive or negative health outcomes in the future [41].

Alcohol's associations with comfort, reward and relaxation had unsurprising appeal for those managing the discomfort and uncertainty of living with chronic health problems. The unpleasant biological/neurological effects of alcohol in terms of hangovers were well known, but these down-sides were normalised, i.e. hangovers were regarded as commonplace and so not particularly threatening. In contrast to medicines (and indeed smoking), there was little attention paid to the risk of alcohol contributing to ill health over time except, for others deemed susceptible, in terms of dependence. There was something of a paradox for interviewees relying on evidence of alcohol harms from tests or feeling out of control as spurs to change, which would only become evident after some health damage had been done.

Muhlack et al (2018) state that little is known about the motivations and decision-making processes of "non-problematised" middle aged drinkers in a recent qualitative systematic review [25]. The review found that health was not identified as a significant consideration in this population, except where drinking behaviours were likely to harm others. Based on these

results, they suggest that public health campaigns might have more impact if focused on unacceptable drinking behaviours instead of personal health outcomes. However, interviewees in our study with reason to be conscious of health messages, seemed to be already responsive to some warnings, and not others. Most did not see susceptibility to harmful drinking in themselves and seemed largely unaware of the direct risks of alcohol to their own long-term health. People were mostly drinking at lower levels than at previous stages of their lives, but nonetheless still drinking at risky levels. They judged these amounts as moderate when contrasted with the drinking done by their former selves or others who were perceived to be drinking more [40]. The association of problematic drinking with lack of personal control rather than potential health harms here and in other studies [22] [42], is arguably congruent with some current public health and all alcohol industry messaging focused on 'responsibility' in drinking.

Risk communication can be defined as, "the open two way exchange of information and opinion about harms and benefits" [43]. This study points to the need for better knowledge of how and why people understand risks that relate to alcohol, the temporal construction of such risks, and the implications for the management of long-term health conditions. This includes the effects of the broad social influences on determinants of drinking behaviour and what is regarded as normalised and acceptable, and the processes by which this occurs. Such factors contribute to health and ill health, and also have implications for the quality and applicability of alcohol advice given in relation to the management of long-term health conditions.

People continue to drink while taking multiple medications, despite the advice they have been given by health care professionals, which appears to be largely disregarded except where supportive of drinking. Alcohol guidelines are aimed at the general population, which can appear rather abstract, and there is evidence to suggest that health professionals who could help personalise these guidelines are not always comfortable or confident in talking about alcohol [44–46]. Something different is needed to support patients with long-term conditions to understand the potential risks of alcohol, and to think through the implications for their health and well-being over the long term. Findings from this study and other work in the CHAMP-1 programme are being used to inform the development of a novel intervention to help raise the subject of alcohol in medication reviews for those taking medications long-term, and to better manage consultations in a person-centred manner.

These interviews are rich accounts but they are generated in a particular (research) setting, and are therefore in part framed by the interviewer and the wider project (focused on medication, drinking and a potential pharmacy intervention). In keeping with qualitative work, the study makes no claims for generalisability but aims to provide a level of detail and transparency to allow readers to determine the extent of potential transferability to other contexts. While the sample size is sufficient to describe a range of views in a specific context, it lacks diversity in terms of ethnicity and sexual identity and includes fewer women than men. It includes a wide range of combinations of drinking behaviours, medication use and conditions and provides novel data on how drinking and medicines are used together.

## Conclusion

In order to help prevent health harms arising from alcohol consumption, health risk communication should be informed by high quality evidence, including evidence about how people with long-term health conditions perceive health outcomes over time, and how they manage their concurrent alcohol and medicines use. There is thus a need for underpinning evidence on both the nature of the risk in different populations, and on the processes of communication. Interventions concerned with medicines adherence and optimisation should include consideration of the concurrent use of the legal and widely available psychoactive drug alcohol.

Health risk communication should not be conceptualised as a one-way process of imparting information. Discussions about risks could better support people to make active and informed connections between medicines, alcohol and their health if they are perceived as promoting rather than restricting personal choice and autonomy.

## Supporting information

**S1 Appendix. Types of prescribed medication taken by interviewees.**
(DOCX)

## Acknowledgments

Disclaimer: The views expressed are those of the authors and not necessarily those of the NIHR or the Department of Health and Social Care.

Thanks to the interviewees and the patient and public involvement group who advised on the study. This research was funded by the National Institute for Health Research (NIHR) Programme Grants for Applied Research (RP-PG-0216-20002).

## Author Contributions

**Conceptualization:** Mary Madden, Duncan Stewart, Karl Atkin, Brendan Gough, Jim McCambridge.

**Formal analysis:** Mary Madden.

**Investigation:** Mary Madden, Stephanie Morris.

**Methodology:** Mary Madden, Karl Atkin, Brendan Gough, Jim McCambridge.

**Writing – original draft:** Mary Madden.

**Writing – review & editing:** Mary Madden, Stephanie Morris, Duncan Stewart, Karl Atkin, Brendan Gough, Jim McCambridge.

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
