## [Decision Letter · Decision Letter 0]

8 Aug 2019

PONE-D-19-19395

Conceptualising alcohol consumption in relation to long term health conditions: exploring risk in interviewee accounts of drinking and taking medication

PLOS ONE

Dear Dr Madden,

Thank you for submitting your manuscript to PLOS ONE. After careful consideration, we feel that it has merit but does not fully meet PLOS ONE’s publication criteria as it currently stands. Therefore, we invite you to submit a revised version of the manuscript that addresses the points raised during the review process.

We would appreciate receiving your revised manuscript by Sep 22 2019 11:59PM. To enhance the reproducibility of your results, we recommend that if applicable you deposit your laboratory protocols in protocols.io, where a protocol can be assigned its own identifier (DOI) such that it can be cited independently in the future. For instructions see: http://journals.plos.org/plosone/s/submission-guidelines#loc-laboratory-protocols

We look forward to receiving your revised manuscript.

Kind regards,

Joel Msafiri Francis, MD, MS, PhD

Academic Editor

PLOS ONE

Journal Requirements:

1. We note that you have indicated that data from this study are available upon request. PLOS only allows data to be available upon request if there are legal or ethical restrictions on sharing data publicly. For information on unacceptable data access restrictions, please see http://journals.plos.org/plosone/s/data-availability#loc-unacceptable-data-access-restrictions.

Reviewers' comments:

Reviewer's Responses to Questions

**Comments to the Author**

1. Is the manuscript technically sound, and do the data support the conclusions?

Reviewer #1: Yes

Reviewer #2: Yes

2. Has the statistical analysis been performed appropriately and rigorously? 

Reviewer #1: N/A

Reviewer #2: N/A

3. Have the authors made all data underlying the findings in their manuscript fully available?

Reviewer #1: Yes

Reviewer #2: No

4. Is the manuscript presented in an intelligible fashion and written in standard English?

Reviewer #1: Yes

Reviewer #2: Yes

5. Review Comments to the Author

Reviewer #1: The present paper describes a collection of qualitative interviews with individuals whom drink alcohol recreationally and are prescribed medications for long-term illness, with the aim of further exploring how these individuals understand their alcohol use. This is a relevant and timely topic, with concurrent alcohol and prescription medication use causing significant issues in relation to health outcomes globally. As noted by the authors, qualitative interviews are a good opportunity to explore these issues in-depth. However, I would not recommend this paper for publication without some key changes.

Specifically, the structure of the paper (most notably the ‘results’ section) could be refined to improve flow and reader comprehension. I would recommend first explicitly outlining the nature of alcohol use amongst the sample (regarding AUDIT scores, and what this means), then discussing the types of medications being used by interviewees, and finally how these substances exist together.

Descriptive statistics (Table 1 and Table 2) would heavily benefit from some expansion (see comments below).

Some sections of the manuscript are not clear. At the beginning of each section, I would recommend specifically outlining what the section is about to cover, and in a way that a lay individual would understand.

The health intervention implications described in the manuscript could be expanded upon.

Please find specific comments below.

Abstract:

“Risky drinking was located elsewhere”: The meaning of this sentence isn’t immediately clear without having first read the body of the text.

Introduction:

“experience and understand their use of alcohol, the psychoactive drug ethanol”: Even though it’s fairly obvious which compound is being discussed, if the authors wish to make the clarification I would recommend stating that “ethanol” is the alcohol of focus early in the introduction rather than at the end.

Methods:

“The interview schedule and recruitment materials were both developed with the input of lay advisors.”: It would help to specify why this was done.

“For example, INV-015-M-24-05 means that 119 interviewee number 15 is male with an AUDIT-C score of 5 and a MARS score of 24.”: I would recommend presenting the explanation of these codes respective to their placement within the code. i.e., participant – gender – MARS score – AUDIT score

Please include an explanation of measure interpretation (AUDIT and MARS) alongside first description of the measures. Further, it would help to more explicitly explain why these were being collected and their importance in contextualizing interviewee responses.

The introduction made explicit reference to older individuals as a susceptible population, but the methods section describes broad demographic selection criteria. Is there a reason why older individuals weren’t specifically targeted? This could use some clarification.

Results:

3.1:

Table 1: I would highly recommend including, at minimum, standard deviations for all descriptive statistics. There’s also inconsistent punctuation throughout this table.

3.2:

Table 2: As above, standard deviations should be included.

The appendix noted in text has not been provided. Also, if possible, it would help to refer to specific appendices (e.g., Appendix A) unless they are provided as broader supplementary material.

“Many interviewees had AUDIT-C scores ranging 142 between 5 and 10”: How many interviewees specifically? Perhaps the n of AUDIT “groups” could be included in one of the tables?

I would recommend restructuring this section to improve clarity. Perhaps you could first discuss alcohol, then medicines/illicit substances (or vice versa), and finally move on to related practices. The structure as it currently stands seems slightly haphazard.

3.4:

“Locating risky drinking elsewhere”: This title would benefit from some clarification. Is this section about the identification of risky drinking practices amongst peers?

Discussion:

“A less one way style of risk communication could be the start of better supporting people to make active and informed connections between medicines, alcohol and potential personal health harms if this is to be perceived as promoting rather than restricting agency/choice/autonomy.”: I think this is a great takeaway from these data, but it would be nice to see an expansion on the types of strategies that could be implemented and the contexts they would exist in. Would this strictly apply to health professionals in a health service context? Or could it be applied more broadly? How could we further research in this area?

Reviewer #2: This is a very interesting and important manuscript describing the qualitative findings of interviews conducted with persons taking in North England who take medications for chronic conditions and who are current alcohol users (at least 2 days per week). The topic is very important because health guidelines are given, but not well understood or followed, and alcohol use is a highly prevalent social and self-medicating activity. The findings of the balance of short-term pleasure versus long-term harm and the concept of drinking problems being those of others were particularly illustrative.

The manuscript is well-written; suggested revisions are as follows.

Please give more detail on how participants were recruited. How were they identified as taking alcohol at least 2 times per week, and what qualified as “eligible for medication reviews”. After being deemed potentially eligible, how were they approached, and asked to participate? Was the recruitment done on site at the pharmacies, and were there any particular hours of the day? How many declined participation and were there any patterns (age, sex, health)?

Please provide a description of standard cutoffs for hazardous drinking (via AUDIT-C) and poor adherence (MARS) with references, for those unfamiliar with these scales.

The tables should be edited to be in standard format – so that variable names and categories on are the left and numbers on the right. Percentages should be given for categorical variables in addition to the n.

Table 1 has n=25, Table 2 has n=24, removing the one person removed from the analyses. I suggest describing that person’s removal in the 1st paragraph of results (along with the suggested data about recruitment and participation). Then present results in the text and tables for n=24.

There are a few places where there is a reference to a person, and it is unclear if it refers to the following or preceding person being quoted. Please clarify.

Section 3.2.2 is focused on medication side effects, regardless of drinking (except at the very end) and could be dropped.

There is also reference to the participants’ past drinking in the Discussion, that is mentioned only in a few of the results in section 3.5. Please provide these results.

The recommendations in the discussion are very thin. The authors cite others’ recommendations (Muhlack et al) but do not say whether those are relevant, and their only recommendations are to obtain better knowledge, and state that “something different is needed”. The one recommendation given, that of a less one-way communication style, is not supported by the findings. What about leveraging the strategies already in place (e.g. drinking water, comparing to others’ drinking)?

It would be preferable to have a final conclusion rather than end with study limitations.

6. PLOS authors have the option to publish the peer review history of their article (what does this mean?). If published, this will include your full peer review and any attached files.

Reviewer #1: No

Reviewer #2: Yes: Judith A. Hahn

---

## [Author Response · Author response to Decision Letter 0]

1 Oct 2019

Response to Reviewers: PONE-D-19-19395

Conceptualising alcohol consumption in relation to long term health conditions: exploring risk in interviewee accounts of drinking and taking medications. PLOS ONE

Thank you to the reviewers for their helpful comments. We have redrafted the paper to improve structure, provide more clarity and make the discussion more substantive. We have taken the opportunity to include some relevant new references including Ahmed on health professionals and risk communication, Khadjesari and colleagues’ recent study on primary care patients’ views on their own drinking and a recent report from Public Health England on dependence and withdrawal associated with some prescribed medicines. We have made minor proofing edits throughout. Responses to points raised by individual reviewers are below. 

Reviewer #1

The present paper describes a collection of qualitative interviews with individuals whom drink alcohol recreationally and are prescribed medications for long-term illness, with the aim of further exploring how these individuals understand their alcohol use. This is a relevant and timely topic, with concurrent alcohol and prescription medication use causing significant issues in relation to health outcomes globally. As noted by the authors, qualitative interviews are a good opportunity to explore these issues in-depth. However, I would not recommend this paper for publication without some key changes.

Specifically, the structure of the paper (most notably the ‘results’ section) could be refined to improve flow and reader comprehension. I would recommend first explicitly outlining the nature of alcohol use amongst the sample (regarding AUDIT scores, and what this means), then discussing the types of medications being used by interviewees, and finally how these substances exist together.

Some sections of the manuscript are not clear. At the beginning of each section, I would recommend specifically outlining what the section is about to cover, and in a way that a lay individual would understand.

• The manuscript has been amended to improve structure and flow incorporating the specific suggestions above (detail below)

The health intervention implications described in the manuscript could be expanded upon.

• These have been expanded (see below)

Abstract:

“Risky drinking was located elsewhere”: The meaning of this sentence isn’t immediately clear without having first read the body of the text.

• We have amended this in the abstract and the results to aid clarity. The abstract now reads ‘Risky drinking was located elsewhere, in the drinking of others’ L28. We have also amended heading 3.4 ‘Locating risky drinking in the drinking of others’ L435.

Introduction:

“experience and understand their use of alcohol, the psychoactive drug ethanol”: Even though it’s fairly obvious which compound is being discussed, if the authors wish to make the clarification I would recommend stating that “ethanol” is the alcohol of focus early in the introduction rather than at the end.

• We have introduced this earlier as suggested, in the first paragraph of the introduction L.45-46.

Methods:

“The interview schedule and recruitment materials were both developed with the input of lay advisors.”: It would help to specify why this was done.

• Patient and public involvement in research is expected by the funders of the programme, the National Institute of Health Research. We report on our approach to this elsewhere. Rather than get into the detail here we have inserted, “…as part of the research programme commitment to co-production and patient and public involvement (reported elsewhere).” L102-104

“For example, INV-015-M-24-05 means that 119 interviewee number 15 is male with an AUDIT-C score of 5 and a MARS score of 24.”: I would recommend presenting the explanation of these codes respective to their placement within the code. i.e., participant – gender – MARS score – AUDIT score

• Thanks for noticing this, the explanation of the codes has been amended as suggested L154-155

Please include an explanation of measure interpretation (AUDIT and MARS) alongside first description of the measures. Further, it would help to more explicitly explain why these were being collected and their importance in contextualizing interviewee responses.

• Detail clarifying the use of AUDIT and MARS measures has been added to the methods section L131-138 

The introduction made explicit reference to older individuals as a susceptible population, but the methods section describes broad demographic selection criteria. Is there a reason why older individuals weren’t specifically targeted? This could use some clarification.

• The focus of the study is on people with long term conditions rather than age per se. We have provided more detail on our recruitment methods for clarification. As noted in the introduction, the risk of LTCs and the likelihood of taking more medicines increases with age. The sample achieved ranged in age from 41-89 and findings from studies on older and middle aged drinkers and alcohol-medication interactions in older age groups provide relevant context. We have attempted to clarify this and have deleted the phrase ‘older adults’ in the introduction L55 in case this leads to the assumption that (as in some of the studies cited) that age is our primary focus. 

Results:

Descriptive statistics (Table 1 and Table 2) would heavily benefit from some expansion 

3.1: Table 1: I would highly recommend including, at minimum, standard deviations for all descriptive statistics. There’s also inconsistent punctuation throughout this table.

3.2: Table 2: As above, standard deviations should be included

•Table 1 is now a list of types of pharmacies involved in the study in response to reviewer 2’s comments below. Table 2 presents characteristics of the interview sample. This has been reformatted and includes standard deviations. Additional numerical data has been included in the text e.g. percentages of respondents. 

The appendix noted in text has not been provided. Also, if possible, it would help to refer to specific appendices (e.g., Appendix A) unless they are provided as broader supplementary material.

• S1 Appendix: ‘Types of prescribed medication taken by interviewees’ now provided

 “Many interviewees had AUDIT-C scores ranging 142 between 5 and 10”: How many interviewees specifically? Perhaps the n of AUDIT “groups” could be included in one of the tables?

• Specific scores have been reported for AUDIT-C see L172-174. 

I would recommend restructuring this section to improve clarity. Perhaps you could first discuss alcohol, then medicines/illicit substances (or vice versa), and finally move on to related practices. The structure as it currently stands seems slightly haphazard.

• This section has been restructured for clarity starting with alcohol and moving on to medicines use as suggested L171-213

3.4: “Locating risky drinking elsewhere”: This title would benefit from some clarification. Is this section about the identification of risky drinking practices amongst peers?

• We have amended this in the abstract and the results to aid clarity. The abstract now reads ‘Risky drinking was located elsewhere, in the drinking of others’ L28. We have also amended heading 3.4 ‘Locating risky drinking in the drinking of others’ L435.

Discussion:

“A less one way style of risk communication could be the start of better supporting people to make active and informed connections between medicines, alcohol and potential personal health harms if this is to be perceived as promoting rather than restricting agency/choice/autonomy.”: I think this is a great takeaway from these data, but it would be nice to see an expansion on the types of strategies that could be implemented and the contexts they would exist in. Would this strictly apply to health professionals in a health service context? Or could it be applied more broadly? How could we further research in this area?

• The discussion has been expanded. It now includes more on communicating risk and an example of how these findings are being used for intervention development. 

Reviewer #2

This is a very interesting and important manuscript describing the qualitative findings of interviews conducted with persons taking in North England who take medications for chronic conditions and who are current alcohol users (at least 2 days per week). The topic is very important because health guidelines are given, but not well understood or followed, and alcohol use is a highly prevalent social and self-medicating activity. The findings of the balance of short-term pleasure versus long-term harm and the concept of drinking problems being those of others were particularly illustrative.

The manuscript is well-written; suggested revisions are as follows.

• Thank you

Please give more detail on how participants were recruited. How were they identified as taking alcohol at least 2 times per week, and what qualified as “eligible for medication reviews”. After being deemed potentially eligible, how were they approached, and asked to participate? Was the recruitment done on site at the pharmacies, and were there any particular hours of the day? How many declined participation and were there any patterns (age, sex, health)?

• We have added detail on the recruitment process including details of the pharmacies in which recruitment took place L104-143

Please provide a description of standard cutoffs for hazardous drinking (via AUDIT-C) and poor adherence (MARS) with references, for those unfamiliar with these scales.

• Detail clarifying the use and interpretation of AUDIT and MARS measures has been added to the methods section L131-138 and throughout. We hope that this clarifies that drinking above the threshold AUDIT-C score of 4/3 for men and women respectively is in some way hazardous

The tables should be edited to be in standard format – so that variable names and categories on are the left and numbers on the right. Percentages should be given for categorical variables in addition to the n.

Table 1 has n=25, Table 2 has n=24, removing the one person removed from the analyses. I suggest describing that person’s removal in the 1st paragraph of results (along with the suggested data about recruitment and participation). Then present results in the text and tables for n=24.

• Table 1 is now a list of types of pharmacies involved in the study in response to reviewer 2’s comments below. Table 2 presents characteristics of the interview sample. This has been edited and amended. Percentages have been included where numbers of responses are mentioned throughout. We have removed the person not included in this study from the tables and explained this earlier as suggested. The abstract has also been amended [n=24]

There are a few places where there is a reference to a person, and it is unclear if it refers to the following or preceding person being quoted. Please clarify.

• We have made amendments to clarify this throughout. 

Section 3.2.2 is focused on medication side effects, regardless of drinking (except at the very end) and could be dropped.

• The relevance of this section has been clarified by introducing some explanatory text on the contrast in views on medicines and alcohol (as drugs) in relation to a sense of control or autonomy. The section has been re-named L263-269

There is also reference to the participants’ past drinking in the Discussion, that is mentioned only in a few of the results in section 3.5. Please provide these results.

• We have included results on levels of past drinking in comparison with present drinking for the full sample L175-178

The recommendations in the discussion are very thin. The authors cite others’ recommendations (Muhlack et al) but do not say whether those are relevant, and their only recommendations are to obtain better knowledge, and state that “something different is needed”. The one recommendation given, that of a less one-way communication style, is not supported by the findings. What about leveraging the strategies already in place (e.g. drinking water, comparing to others’ drinking)?

• The discussion has been expanded. It now includes a reference to Ahmed H, Naik G, Willoughby H, Edwards AG. Communicating risk. BMJ. 2012;344:e3996 and to literature on health professionals perceptions about talking about alcohol. In addition to recommendations for further study it includes an explanation of how these findings are being used for intervention development in the context of pharmacy. 

It would be preferable to have a final conclusion rather than end with study limitations.

• A conclusion has been added which makes clear our recommendations

---

## [Decision Letter · Decision Letter 1]

21 Oct 2019

Conceptualising alcohol consumption in relation to long term health conditions: exploring risk in interviewee accounts of drinking and taking medications

PONE-D-19-19395R1

Dear Dr. Madden,

We are pleased to inform you that your manuscript has been judged scientifically suitable for publication and will be formally accepted for publication once it complies with all outstanding technical requirements.

With kind regards,

Joel Msafiri Francis, MD, MS, PhD

Academic Editor

PLOS ONE

Additional Editor Comments (optional):

Reviewers' comments:

Reviewer's Responses to Questions

**Comments to the Author**

1. If the authors have adequately addressed your comments raised in a previous round of review and you feel that this manuscript is now acceptable for publication, you may indicate that here to bypass the “Comments to the Author” section, enter your conflict of interest statement in the “Confidential to Editor” section, and submit your "Accept" recommendation.

Reviewer #1: All comments have been addressed

Reviewer #2: All comments have been addressed

2. Is the manuscript technically sound, and do the data support the conclusions?

Reviewer #1: Yes

Reviewer #2: Yes

3. Has the statistical analysis been performed appropriately and rigorously? 

Reviewer #1: (No Response)

Reviewer #2: Yes

4. Have the authors made all data underlying the findings in their manuscript fully available?

Reviewer #1: No

Reviewer #2: No

5. Is the manuscript presented in an intelligible fashion and written in standard English?

Reviewer #1: Yes

Reviewer #2: Yes

6. Review Comments to the Author

Reviewer #1: The authors have done a good job at actioning previous comments. I note that the authors are not making data from the manuscript available, but assume they have made a case to the editor for this.

Reviewer #2: (No Response)

7. PLOS authors have the option to publish the peer review history of their article (what does this mean?). If published, this will include your full peer review and any attached files.

Reviewer #1: No

Reviewer #2: Yes: Judith Hahn

---

## [Editor Report · Acceptance letter]

29 Oct 2019

PONE-D-19-19395R1 

Conceptualising alcohol consumption in relation to long-term health conditions: exploring risk in interviewee accounts of drinking and taking medications 

Dear Dr. Madden:

I am pleased to inform you that your manuscript has been deemed suitable for publication in PLOS ONE. Congratulations! Your manuscript is now with our production department. 

With kind regards,

on behalf of

Dr. Joel Msafiri Francis 

Academic Editor

PLOS ONE